# On the Effect of Model Parameters on Forecast Objects

Caren Marzban[1,2]*, Corinne Jones[2], Ning Li[2], Scott Sandgathe[1]

[1] Applied Physics Laboratory

[2] Department of Statistics

Univ. of Washington, Seattle, WA 98195 USA

ABSTRACT

Many physics-based numerical models produce a gridded, spatial field of forecasts, e.g., a temperature "map." The field for some quantities generally consists of spatially coherent and disconnected "objects." Such objects arise in many problems, including precipitation forecasts in atmospheric models, Eddy currents in ocean models, and models of forest fires. Certain features of these objects (e.g., location, size, intensity, and shape) are generally of interest. Here, a methodology is developed for assessing the impact of model parameters on features of forecast objects. The main ingredients of the methodology include the use of 1) Latin hypercube sampling for varying the values of the model parameters, 2) statistical clustering algorithms for identifying objects, 3) multivariate multiple regression for assessing the impact of multiple model parameters on the distribution (across the forecast domain) of

---
*Corresponding Author: marzban@stat.washington.edu

object features, and 4) methods for reducing the number of hypothesis tests, and controlling

the resulting errors. The final "output" of the methodology is a series of boxplots and

confidence intervals that visually display the sensitivities. The methodology is demonstrated

on precipitation forecasts from a mesoscale numerical weather prediction model.

# 1. Introduction

Complex, physics-based numerical models of natural phenomena often have parameters - henceforth, model parameters - whose values are generally not *a priori* specified. In such situations it is important to infer the manner in which the model parameters affect the outputs of the model (i.e., forecasts, or predictions), and often the techniques of Sensitivity Analysis (SA) are employed to assess the effects. There is a wide range of techniques from relatively simple one-at-a-time method (also known as the Morris method) where each model parameter is varied individually (e.g., Yu et al. (2013)), to multivariate approaches motivated by statistical methods of experimental design (Montgomery 2009) where the values of the model parameters are varied according to some optimization criterion. Alternative approaches can be found in Backman et al. (2017) where algorithmic differentiation is used, and in Kalra et al. (2017) where the underlying physics equations are integrated using quadrature methods. And yet another alternative is the adjoint method, commonly used in meteorological circles (Errico 1997).

It is difficult to classify the various methods into a simple taxonomy (Bolado-Lavin and Badea 2008), but the terms Local and Global have been used to denote two broad categories (Saltelli et al. 2010, 2008); generally, local methods employ some sort of derivative of the model output with respect to inputs, while global techniques rely on a decomposition of the variance of the output in terms of the variance explained by the inputs. Comparisons of the various approaches are not common-place, because each approach is usually suited for a specific application where other methods may not be practically feasible. However, an example of the comparison of one global approach and one local (adjoint) approach on the

Lorenz '63 model (Lorenz 1963) has been performed by Marzban (2013).

Another possible classification criterion is based on the purpose of the SA. Some SA work is performed for assessing how model parameters impact the model itself, not as a means to some other goal. For example, Lucas et al. (2013) uses a global SA method to explore the effect of model parameters on the probability of model crashes. By contrast, sometimes SA is performed as an intermediate step to another goal, such as the calibration of the model (Safta et al. 2015; Hacker et al. 2011; Laine et al. 2012; Ollinaho et al. 2014). All of these classification criteria are imperfect, as there exist works which fall "between" Global versus Local, or SA-only versus SA-for-calibration; some examples include Roebber (1989); Roebber and Bosart (1989); Robock et al. (2003). The work reported here falls into the Local and SA-only category; as such, although the proposed methodology can be used for calibration, no attempt is made to do so here.

In many SA studies, the output of the model (i.e., the response variable in the SA) is usually a single or a handful of scalar quantities. But there are situations in which the output is a gridded spatial field, e.g., temperature forecasts over a spatial region. Every grid point reflects a forecast at that location, and for a quantity like temperature the field as a whole has a smooth, continuous nature. SA is more complicated for precipitation fields, where the model output is a quantity whose spatial structure is not smooth and/or continuous. Indeed, there may be a coherent set of grid points that receive no precipitation at all, while an adjacent set of grid points will reflect a complex pattern of precipitation. In short, the spatial field of such quantities will contain "objects" within which precipitation does occur, surrounded by regions of little or no precipitation. Such objects arise in a wide range of Earth systems, e.g., models of ocean currents and eddies (e.g., Fig. 1 in Samsel et al.

₆₄ (2015)), atmospheric plume/dispersion (e.g., Fig. 4 in Stein et al. (2015)), ocean garbage

₆₅ transport (e.g., Fig. 2 in Froyland et al. (2014)), forest fires (e.g., Fig. 8 in Vogelmann et al.

₆₆ (2011)), and models of the Earth's mantle (e.g., Fig. 4. in French et al. (2013)).

₆₇ For such discrete fields, the assessment of the quality of the forecasts has given rise to a

₆₈ wide range of specialized techniques generally referred to as spatial verification (or evalua-

₆₉ tion) (Ahijevych et al. 2009; Baldwin et al. 2001, 2002; Brown et al. 2002; Casati et al. 2004;

₇₀ Davis et al. 2006a,b; Du and Mullen 2000; Ebert 2008; Ebert and McBride 2000; Gilleland

₇₁ et al. 2009; Hoffman et al. 1995; Keil and Craig 2007; Marzban and Sandgathe 2006, 2008;

₇₂ Marzban et al. 2008, 2009; Nachamkin 2004; Roberts and Lean 2008; Wealands et al. 2005;

₇₃ Wernli et al. 2008; Venugopal et al. 2005; Li et al. 2015). A subset of these methods employs

₇₄ the notion of an object explicitly. In some applications, the object is defined subjectively

₇₅ - for example, by human experts. In other applications statistical methods for clustering

₇₆ (Everitt 1980) are used to identify/define objects within the field (Marzban and Sandgathe

₇₇ 2006, 2008). This clustering approach, which has been re-examined by Lakshmanan and

₇₈ Kain (2010), and more recently by Wang et al. (2015), is the basis of the object-identification

₇₉ procedure used in the present work.

₈₀ Although no spatial verification/evaluation is done here, the importance of objects within

₈₁ the forecast field calls for a SA framework wherein one can assess the effect of model param-

₈₂ eters on features of the objects. Also, the assessment of sensitivity is highly intertwined with

₈₃ that of statistical significance. The methodology developed here can be viewed as an object-

₈₄ based SA with which one can assess the impact (both the magnitude and the statistical

₈₅ significance) of model parameters on object features.

₈₆ More specifically, the next section describes the main components of the proposed

methodology, namely Latin hypercube sampling for determining how the model parameters are varied (section 2a), and use of clustering algorithms for identifying objects in the forecast field (section 2b). The object features examined here, generally of interest in many applications, include size, location, intensity, and shape, all of which can be readily estimated from the forecasts directly (section 2c). Section 2d describes multivariate multiple regression for assessing the impact of the model parameters on the distribution (across the forecast domain) of object features. Anticipating the problems associated with multiple hypothesis testing, steps are taken to first reduce the number of tests, and then to control different error rates (section 2e). Ultimately boxplots and confidence intervals are used to visually display the daily variability of the sensitivities. Section 2f summarizes all of these components, and is followed by a demonstration of the methodology on forecasts from a weather prediction model (section 3). The paper ends with a statement of the conclusions, additional discussion, and ways in which the methodology can be generalized (section 4).

# 2. Method

*a. Data*

The numerical model employed to demonstrate the methodology is COAMPS$^{\circledR}$ (Hodur 1997), for which some SA work has already been done. Doyle et al. (2011) and Jiang and Doyle (2009) examine the effect of model parameters on mountain waves. Motivated by the work of Holt et al. (2011) who studied the effect of 11 model parameters on various characteristics of the forecasts, Marzban et al. (2014) used a global, variance-based SA to

study the effect of the same parameters and their interactions on mean (across the forecast domain) and the center-of-gravity of precipitation. By contrast, here, the effect of the model parameters is assessed on features of objects within the forecast field. As discussed in section 2c, a total of six features are examined, together summarizing the location, intensity, and the shape of each object.

These 11 parameters are the inputs to the numerical model, and the outputs are forecasts of precipitation at each of $45 \times 72$ grid points, with a spacing of $81km$, covering the entire continental US, including coastal regions, and portions of Canada and Mexico. The SA method developed here requires data - technically, *computer data* - which are created by generating an ensemble (or sample) of input values, assimilating surface observations, and then running the model forward to produce 24h forecasts of precipitation amount at each grid point. As such, the SA results are contingent on the nature of this data, and consequently, care must be taken in the data-generation step of the methodology.

The data used for the SA must be representative of the range of phenomena observed at large. To that end, the present application involves a wide range of weather phenomena, spanning 120 days from February 16 through July 2, 2009. Confirmed by visual examination of all 120 forecasts, this temporal period includes a comprehensive series of midaltitude synoptic systems traveling across the northern portion of the domain. These synoptic systems extend down into the southeastern US early in the period and are replaced by subtropical convective systems in the late spring and summer months. This subtropical activity also occurs in the southwestern portion of the domain (west coast of Mexico) during June and July in association with the southwest monsoon. The only apparent atypical weather appears to be a greater amount of convective activity off the east coast of the US associated with

<sup>130</sup> quasi-stationary or slow moving frontal systems during the period.

<sup>131</sup> It is important that the data cases are as independent as possible. To that end, the 120

<sup>132</sup> days are sampled at 3-day intervals in order to minimize temporal dependency, leading to

<sup>133</sup> 40 days for the analysis.

<sup>134</sup> For each of the 40 days, 99 different values for 11 parameters are generated by Latin

<sup>135</sup> Hypercube Sampling (LHS). Said differently, for each day, a sample of size 99 is taken from

<sup>136</sup> the 11-dimensional space of the model parameters. This so-called "space-filling" sampling

<sup>137</sup> scheme assures that no two of the 99 points have the same value for any of the 11 parameters.

<sup>138</sup> It can be shown that this property leads to more precise estimates (at least, no less-precise

<sup>139</sup> estimates) than many other sampling schemes (Cioppa and Lucas 2007; Montgomery 2009;

<sup>140</sup> Marzban 2013). LHS is appropriate when the model parameters are all continuous quantities

<sup>141</sup> (i.e., taking values on the Real line). For discrete or categorical inputs, Latin Square Designs

<sup>142</sup> or Fractional Factorial Designs can be employed to produce optimal samples (Montgomery

<sup>143</sup> 2009); these methods will be demonstrated in a separate article.

<sup>144</sup> Given that daily variability is a common source of variability in models dealing with

<sup>145</sup> Earth systems, one question that arises is whether one should use a given LHS sample for all

<sup>146</sup> days in the analysis. Here, in order to explore a larger portion of the model parameter space,

<sup>147</sup> the LHS sample is allowed to vary across each of the 40 days in the study. Although this

<sup>148</sup> choice confounds variability due to model parameters with daily variability, it is arguably a

<sup>149</sup> better choice than the alternative (of using the same LHS sample across all days) because

<sup>150</sup> the final sensitivity results will not be contingent on a given LHS sample.

<sup>151</sup> The 11 model parameters are shown in Table 1; the choice of these parameters is ex-

<sup>152</sup> plained in Holt et al. (2011). As mentioned in that paper, these parameters were chosen

for their anticipated sensitivity (through model tests and discussions with developers) of the parameterizations in an effort to choose parameters most likely to produce changes in the model output precipitation fields. Also, to focus on heavy precipitation, only the grid points whose convective precipitation amount exceeds the 90th percentile of precipitation across the domain are analyzed.

## b. Cluster Analysis

There exists a wide range of clustering methods, each with their respective parameters (Everitt 1980). At one extreme, there exists a class of clustering methods wherein the desired number of cluster, $NC$, is specified by the user. A proven example in this class is called Gaussian Mixture Model (GMM) clustering (McLachlan and Peel 2000). At the other extreme, there exist clustering routines where $NC$ does not play a role at all. One such method is called Density-Based Spatial Clustering of Applications with Noise (DBSCAN) (Ester et al. 1996). DBSCAN has two parameters, here denoted $\epsilon$ and min_samples. Roughly speaking, $\epsilon$ is the maximum distance between two grid points in order for them to be in the same cluster, and min_samples is the minimum number of grid points necessary to form a cluster.

Here, these two approaches are selected for demonstration because they allow for two very different ways in which a user can inject *a priori* knowledge into the analysis. For example, in some applications it may be more natural to specify the number of clusters, in which case GMM is a natural choice. On the other hand, DBSCAN is more natural if the user has knowledge of the typical size and distance between clusters. For example, consider

a situation wherein the grid-spacing is relatively large (as is the case in this paper, i.e., $81km$), allowing one to examine only large scale precipitation. Although time of year and location are also important, if one were to focus only on winter months in, say, the Pacific Northwest, then it is reasonable to set $\epsilon$ to 3 or 4. By contrast, if one is considering jet streaks, e.g., where some maximum wind speed value is reached, then $\epsilon$ can be closer to 1. As for min_samples, 4 or 5 are reasonable values for both precipitation and jet streak events, at the model resolution used here.

In addition to the way in which the respective parameters are handled, another reason why these two clustering methods are used here is that they occupy two other extremes in the family of clustering algorithms: GMM clustering belongs to a class of model-based algorithms (Banfield and Raftery 1993; Fraley and Raftery 2002) common in statistics circles because they are conducive to performing statistical tests, while DBSCAN assumes no underlying model, and for this reason is often employed in machine learning applications.

For the SA component of the methodology developed here, it is not necessary for the objects to be defined by these or any other clustering algorithm; the objects may be defined by any other criterion or even by human experts. But some general guidance on the available options may be in order. As mentioned previously, some algorithms require the specification of the number of clusters (e.g., GMM) while others require information on the desired size and/or distance between clusters (e.g., DBSCAN). There exists another class of clustering algorithms wherein no such specification is required; an example of this type is the hierarchical agglomerative clustering (Everitt 1980), wherein the procedure begins by assigning each of $N$ points to a unique cluster, and then proceeds by combining the clusters systematically until all points are members of a single cluster. As such, this algorithm allows the

number of clusters to vary systematically from $N$ to 1. A variation on this routine involves the reverse procedure wherein the number of clusters is varied from 1 to $N$. The clustering results may depend on the choice of these procedures, and so, for any specific problem some trial-and-error experimentation is recommended.

In clustering algorithms that rely on a notion of distance, there are two types of distance that must be distinguished, generally referred to as intra-cluster and inter-cluster. The former refers to the distance between any two points, while the latter gauges the "distance" or similarity between two clusters. On gridded fields, the notion of an intra-cluster distance is itself ambiguous; two common choices are the Euclidean distance (defined by the Pythagorean theorem), and the Manhattan distance (defined by the sum of the grid lengths connecting two grid points). Although the resulting clusters do depend on the choice of this distance measure, the former generally lead to smaller and more distant clusters. Here, in DBSCAN, the Euclidean intra-cluster distance is used; GMM does not involve the notion of an intra-cluster distance.

In clustering algorithms that involve the notion of an inter-cluster distance, some consideration must be given to at least three common measures: 1) the group-average distance (defined as the average of the intra-cluster distances between all the points across two clusters), 2) the distance between the closest grid points across the two clusters, and 3) the distance between the farthest grid points across the clusters. The last two options are often called SLINK (for Shortest or Single link), and CLINK (for Complete link), respectively. Again, the final clustering results may depend on the choice of this distance, but CLINK generally results in tightly packed, small clusters. By contrast, SLINK leads to long and thin clusters. A comparison of these distance measures in clustering of precipitation forecasts is

performed in Marzban and Sandgathe (2006). GMM and DBSCAN do not employ a notion of inter-cluster distance.

Given that all of the above-mentioned choices may affect the final clustering result, and the fact that the notion of an object is user-dependent, no specific choice is recommended here. A similar philosophy is adopted with respect to the values of the parameters of the clustering algorithms; they may be specified by the user, or varied across a range of values, depending on the specific application. Although there exist statistical criteria that lead to unique values for the parameters, the criteria involve the optimization of some other quantity, e.g., Akaike Information Criterion (AIC) or Bayesian Information Criterion (BIC). As such, the ambiguity in the choice of the clustering algorithm, or the values of their parameters, is simply replaced with the ambiguity of selecting the appropriate criterion. Therefore, again, no attempt is made to optimize the values of the parameters. It is assumed that the user has sufficient information about the underlying physics to either specify the number of physical objects (or a range thereof), or the typical size and distance between physical objects.

*c. Cluster Features*

In spatial verification some of the errors that are of interest include displacement, intensity, size/area, and shape error. The estimation of these errors presumes the ability to compute, respectively, the location, intensity, area, and shape of a cluster. Here, the latitude and longitude of the centroid of a cluster are taken as coordinates of its location; intensity is measured by the median (across the spatial extent of the cluster) of precipitation; and area is measured by the number of grid points in a cluster. The shape of a cluster in GMM is

an ellipse because that is the cross-section (i.e., level-set) of a bivariate Gaussian. Then, the eccentricity and orientation of the semi-major axis of the ellipse are natural for quantifying the shape of clusters. In DBSCAN, clusters are not restricted to have any specific shape. In order to be able to compare the two clustering algorithms, here an elliptical shape is assumed for the clusters, and the eccentricity and orientation are obtained from the first and second eigenvectors of the covariance matrix computed from the coordinates of all the grid points in a given cluster. The length of the semi-major axis is set to the largest eigenvalue. The ability to estimate the shape of the ellipse from the covariance matrix is an important component of the methodology, because the alternative of fitting curves through the edges of clusters is a much more complicated task. This covariance matrix is central to the construction of many other features of potential interest (Bookstein 1991).

In short, the six cluster features examined here are latitude, longitude, intensity, area, orientation, and eccentricity. It is worth reiterating that these quantities can be estimated from the forecast field, directly, without any further modelling of the objects. Also, as explained in the next section, in order to assess how the distribution (across the forecast field) of a given feature is affected by the the model parameters, the former is summarized with three moments - minimum, median, and maximum.

*d.  Statistical Model*

The SA methodology in Marzban et al. (2014) is a variance-based approach which allows one to identify linear or nonlinear relationships between the forecast quantities and the model parameters, and even interactions between the model parameters. As a first approximation,

however, it is sufficient to estimate only the linear (i.e., main) effects, because nonlinear and interaction effects are often much smaller than main effects; see, for example, pages 192, 230, 272, 314, 329 in Montgomery (2009), and pages 33-34 in Li et al. (2006). For this reason a linear regression-based model is adequate. Specifically, the effect of the model parameters is assessed via the least-squares estimate of the regression coefficients $\beta_i$ in

$$y = \alpha + \beta_1 x_1 + \beta_2 x_2 + \cdots + \beta_{11} x_{11} + \delta \ , \tag{1}$$

where $x_i$ denote standardized model parameters, $y$ is some cluster feature, and $\delta$ represents any source of variability in $y$ other than from the model parameters. This linear model is further justified by the results (shown below) because when it is specialized to the case of one cluster (i.e., the entire spatial domain), it reproduces the results of the variance-based approach reported in Marzban et al. (2014).

There exists a realization of Eq. (1) in which the response is vector-valued; the model is called Multivariate Multiple Regression (MMR), wherein Eq. (1) is understood as a vector equation, where $y$, $\alpha$, and $\beta_i$ are all vectors (Fox et al. 2013; DelSole and Yang 2011; Rencher and Christensen 2012). Ideally one could allow each component of the response vector to represent a forecast feature of a given object. However, the number of objects/clusters varies across the 99 values of the parameters and across days in the data. Methods for estimating MMR coefficients when the number of responses is a random variable (varying across cases) are not readily available. Therefore, for each of the six features measuring location, intensity and shape, three summary measures are considered: the minimum, median, and maximum (across the clusters in the domain) of the feature. These three quantities can be thought of as a 3-point summary of the distribution (technically, histogram) of the feature, and they

serve as the three responses in MMR. In short, the statistical model used here is

$$
\begin{pmatrix} y_d^{min} \\ y_d^{med} \\ y_d^{max} \end{pmatrix} = \begin{pmatrix} \alpha_d^{min} \\ \alpha_d^{med} \\ \alpha_d^{max} \end{pmatrix} + \begin{pmatrix} \beta_{1,d}^{min} \\ \beta_{1,d}^{med} \\ \beta_{1,d}^{max} \end{pmatrix} x_{1,d} + \begin{pmatrix} \beta_{2,d}^{min} \\ \beta_{2,d}^{med} \\ \beta_{2,d}^{max} \end{pmatrix} x_{2,d} + \cdots + \begin{pmatrix} \beta_{11,d}^{min} \\ \beta_{11,d}^{med} \\ \beta_{11,d}^{max} \end{pmatrix} x_{11,d} + \begin{pmatrix} \delta_d^{min} \\ \delta_d^{med} \\ \delta_d^{max} \end{pmatrix}
$$
(2)

where min, med, and max denote the minimum, median, and maximum (across clusters),

respectively, and $d = 1, 2, \cdots 40$ days. In this equation, the index corresponding to the

99 samples, across which the regression is performed, has been suppressed. As mentioned

previously, the 99 samples of the 11 model parameters are allowed to vary across the 40 days

- hence the $d$ subscript on the $x's$ in Eq. (2).

In addition to serving as a 3-point summary of the distribution of features, the minimum,

median, and maximum also serve another purpose; the median is useful, because one can

assess the effect of the model parameters on a "typical" cluster; the minimum and maximum

across clusters are useful because they allow one to assess whether a model parameter has

an effect on **any** of the clusters in a field. For example, if it is found that a particular model

parameter is positively (negatively) associated with the minimum (maximum) size across

clusters, then one can conclude that the size of at least one of the clusters in the field is

affected by that parameter. This is an important consideration, because if the size of at least

one of the clusters is not affected by a parameter, then that parameter can be said to have

no effect on the size of clusters.

One may wonder why it is important to use MMR with three responses, as opposed to

three single-response multiple regression models; it is easy to show that the latter ignores

the correlation between the response variables (Fox et al. 2013; Rencher and Christensen

2012). As such, MMR provides a better model of the underlying relationship between the

model parameters and the response variables.

The data on the response variables $y$ are log-transformed to assure more bell-shaped

histograms; this transformation is not necessary, but is useful when the regression coefficients

are subjected to statistical tests, because many such tests assume relatively bell-shaped

distributions.

*e. Significance Tests*

Testing the coefficients in the MMR model involves performing a large number of statis-

tical tests ($40 \times 11 \times 6 \times 3$): one on each of 40 days, for each of 11 parameters, for each of six

cluster features, and for each of three summary measures across clusters. A large number

of tests, in turn, leads to an exponential growth in the probability of making some Type I

error. In general, the increase in the probability of making errors associated with multiple

tests is known as the multiple hypothesis testing problem (Benjamini and Hochberg 1995;

Bretz et al. 2001; Dmitrienko et al. 2009; Montgomery 2009; Rosenblatt 2013; Wilks 2011).

There exist several procedures for addressing this problem, and they all involve two

ingredients: 1) A set of "raw" p-values resulting from multiple hypothesis tests, and 2) the

specification of an error rate to be controlled. Then, the p-values are corrected (usually

scaled) in order to control the error rate. Two common measures of error rate are the

Family-wise Error Rate (FWER), defined as the probability of at least one Type I error, and

the False Discovery Rate (FDR), which is the expected proportion of Type I errors among

all the tests that lead to the rejection of the null hypothesis. One of the simplest procedures

for correcting the p-values involves simply multiplying all of the p-values by the number of tests, and then comparing these corrected p-values with a fixed significance level (e.g. 0.05). This correction controls the FWER, and is called the Bonferroni correction (Bretz et al. 2001; Wilks 2011). One of the popular procedures for controlling the FDR, due to Benjamini and Hochberg (1995), similarly involves scaling each p-value but by a quantity that depends on the rank of the p-value. The choice of the error rate to be controlled is sometimes evident from the nature of the problem (Rosenblatt 2013), but not in the present case; for this reason, both corrections are examined.

Quite independently of the above methods for controlling the errors arising from the multiplicity of tests, there exists a procedure which is often practiced when one is faced with multiple hypothesis tests. The main goal of the procedure is to reduce the number of tests performed, and it is generally possible to do so in tests that involve linear models (Montgomery 2009). In the first stage of the procedure, one performs a single, often-called omnibus, hypothesis test of whether **any** of the predictors (here, model parameters) in the linear model have an effect on **any** of the responses. If the null hypothesis cannot be rejected, then no more tests are performed, and the conclusion of the analysis is that there is no evidence that any of the parameters have an effect on any of the responses. If, however, the null hypothesis is rejected, then, and only then, one proceeds to the second stage of testing the significance of each of the parameters, separately.

In the present application, the omnibus test used in the first stage is called the Pillai's trace test (Fox et al. 2013; Rencher and Christensen 2012), and its use reduces the total number of tests from $(40 \times 11 \times 6 \times 3)$ to only $40 \times 6$. Here, both FWER- and FDR-controlling corrections to these p-values are examined. The second stage of the aforementioned pro-

cedure calls for testing the effect of each of the model parameters separately, but only for those comparisons that have been found significant in the first stage. However, here, for the this second stage, no hypothesis testing is performed at all, because in spite of the plethora of p-values they provide no information on the **magnitude** of the effect of each parameter. Instead, in the second stage, we examine the boxplot of the estimated regression coefficients as well as the associated confidence intervals.

The boxplots are generated and interpreted as follows. For each of the six cluster features, for each of the three summary measures (minimum, median, and maximum across clusters in the whole field), boxplots of the regression coefficients for the 11 model parameters are produced. The degree of overlap between each boxplot and the number zero reflects a visual (though qualitative) assessment of both the statistical significance and the magnitude of the effect of the corresponding model parameter on the response: If zero is well within the span of the boxplot, then one cannot conclude anything regarding the effect; if the boxplot is significantly above (below) zero, then one can conclude that the corresponding parameter has a positive (negative) effect on the response in question; and in such a case, the "distance" of the boxplot relative to zero provides a visual indication of the magnitude of the effect.

The confidence interval for the mean (across 40 days) of the regression coefficient is computed from the estimates of the daily regression coefficients and their standard errors, all computed within MMR. Given that each of the aforementioned displays in the final "output" of the methodology involves 11 CIs, a Bonferroni correction is introduced in order to assure that FWER is maintained at 5%. The interpretation of the CIs is similar to that of the boxplots. If a CI excludes the number zero, one can reject the null hypothesis of no effect with (at least) 95% confidence; otherwise, there is no evidence to draw any conclusion.

The overall position of the CI conveys information on the magnitude of the effect.

A brief discussion of the advantages and disadvantages of the boxplot and the Confidence

Interval (CI) is in order. The boxplot can be considered to provide a 5-point summary of

the empirical sampling distribution of a regression coefficient. The sampling distribution is

more fundamental than the CI (and the p-value) in the sense that the latter is derived from

the former, and as such, the sampling distribution contains more information. However, this

additional information comes at the cost of less rigor, for hypothesis testing with boxplots

is inherently qualitative. CIs introduce a more rigorous display, but they too have some

limitations. For example, whereas hypothesis testing with boxplots does not require a notion

of a confidence level, CIs depend explicitly on that notion. Furthermore analysis of multiple

CIs suffers from the same problems that arise in multiple hypothesis testing with p-values

(see section 2e). Another limitation of CIs is that they are generally symmetric, and so, do

not convey information on the shape (e.g., skew) of the underlying distribution - boxplots

do; see the discussion section for other alternatives. Given the different trade-offs between

boxplots and CIs, both are used here. Consequently, the final output of the methodology

will consist of a figure involving 11 boxplots and CIs (one per model parameter), for each of

six forecast features, and three summary measures (minimum, median, maximum) thereof.

*f. Summary of Method*

This subsection summarizes the main ingredients of the proposed methodology and the

associated problems (and solutions) that arise in an object-based SA. See the flowchart in

Fig. 1.

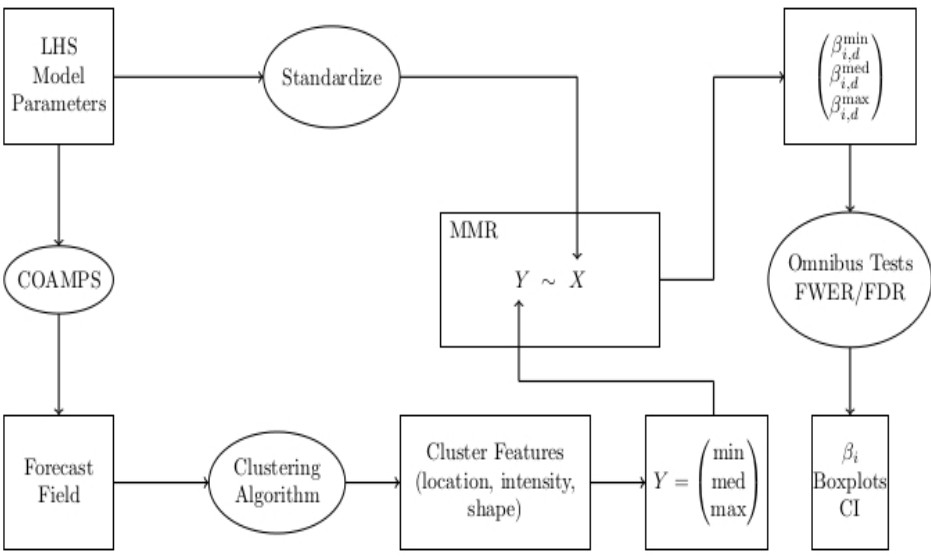

FIG. 1. The flowchart highlighting the main components of the methodology.

In SA, when the model parameters are continuous, a common method for varying them is LHS. It is important to point out that in models wherein daily variability is present, it is advisable to allow the LHS to vary across days.

The model, here COAMPS, is then run for each of the model parameter values in the LHS, and each of the generated forecast fields is subjected to cluster analysis for the purpose of identifying objects in the forecast fields. The choice of the clustering algorithm is an important consideration. Some users may wish to use algorithms in which the number of objects is specified, while other may find it more natural to specify the typical size and/or distance between objects. GMM and DBSCAN are examples from each category. Yet other users may wish to examine all possible clusterings of a field, in which case a hierarchical method is more advisable.

After the objects have been identified, one must decide what object features are of interest. Features that can be estimated directly from the forecast field, without further

modelling, are desirable. The six features proposed here are all readily computed from the forecast field and its spatial covariance matrix.

Given the variability of the object features across the forecast domain, it is then important to assess the effect of the model parameters on the distribution of object features, because the model parameters affect the various objects within a forecast field in different ways. As such, assessing the effect of model parameters on the distribution of features presents a more complete picture of sensitivities than point estimates. Here, a 3-point summary of the distribution is considered: the minimum, median, and maximum.

The question then arises as to how to model the effect of the model parameters on that distribution. Here, it is shown that MMR, with multiple responses corresponding to different moments of the distribution of a features, constitutes an elegant solution. Most notably, MMR allows for omnibus tests of statistical significance which dramatically reduce the number of hypothesis tests. Other steps are also taken to control the error rate associated with multiple hypothesis testing. Then, for each day ($d = 1, \cdots 40$), the MMR coefficients $\beta_{i,d}^{min}, \beta_{i,d}^{med}, \beta_{i,d}^{max}$, with $i = 1, \cdots 11$, provide estimates of the impact of the $i^{th}$ parameter on the distribution of cluster features.

Finally, given the importance of assessing daily variability (at least in the present application), it is proposed that displaying the boxplot of the sensitivities (i.e., the $\beta$'s) across days is more useful than reporting p-values. Such boxplots, although more qualitative than p-values, are more effective in visually displaying both the magnitude and the variability of the sensitivities. Additionally, CIs are also displayed for the purpose of rendering the analysis somewhat less qualitative; see the discussion section for further alternatives.

# 3. Results

As mentioned previously, 24h forecasts are produced for 40 days, each with 99 different values of 11 parameters in COAMPS. Each forecast field is clustered, and three summary measures (minimum, median, and maximum, all across clusters) are computed, each for six cluster features (latitude, longitude, intensity, area, orientation, and eccentricity). First, an omnibus test is performed to test whether any of the 11 parameters have an effect on any of the three summary measures, on each day and for each cluster feature. Then, six MMR models are set up mapping the 11 parameters to three response variables. The daily variability - displayed as boxplots and confidence intervals - for each of the regression coefficients offers a visual assessment of both the statistical significance and the magnitude of the effect of each parameter.[1]

The possibility of performing omnibus tests in MMR reduces the number of tests from $(40 \times 11 \times 6 \times 3)$ to $(40 \times 6) = 240$. The individual p-values are not shown here, but for DBSCAN their histogram is shown in Fig. 2. Evidently, all of the comparisons yield extremely small p-values. At a significance level of 0.05, out of the 240 tests, 53 p-values are not significant when using DBSCAN and 67 are not significant when using GMM. To emphasize the importance of this result, consider the hypothetical situation in which all of these p-values were found to be not significant. In that case, no further hypothesis testing would be necessary at all. Indeed, an examination of the individual p-values displayed in Fig. 2, reveals that a vast majority of the non-significant results are associated with the tests

---

[1]Detailed results on clustering are available; they are suppressed here only to focus on the object-based SA methodology as a whole.

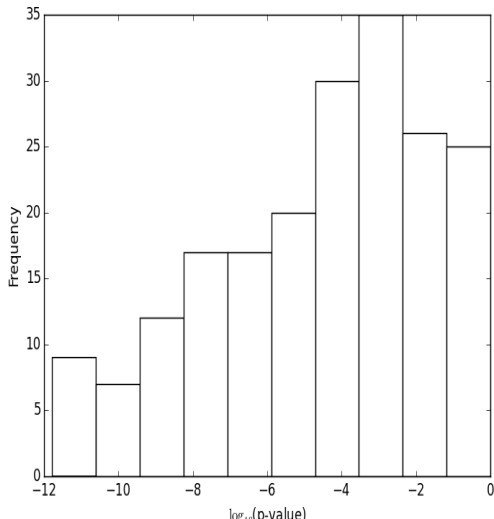

FIG. 2. Histogram of p-values from the omnibus tests across all days and response variables.

when the feature is the eccentricity of an object. As such, one may anticipate that none of the parameters have any effect on eccentricity. The smallness of the remaining p-values, however, calls for proceeding to the second stage of analysis.

The Bonferroni correction for controlling the FWER requires multiplying all of the p-values by the number of tests (i.e., 240). This correction leads to many more nonsignificant comparisons: 129 for DBSCAN and 111 for GMM. Upon making this correction, in addition to eccentricity some of the other features also emerge as being unaffected by any of the 11 parameters. Further details of these results are presented below. When the Benjamini and Hochberg (1995) procedure is applied to control FDR, the number of nonsignificant comparisons is similar to those from the uncorrected tests, i.e., 60 for DBSAN and 74 for GMM.

As mentioned previously, although these rigorous considerations based on p-values are important to assure that the number of false alarms is tamed, it is equally useful to examine

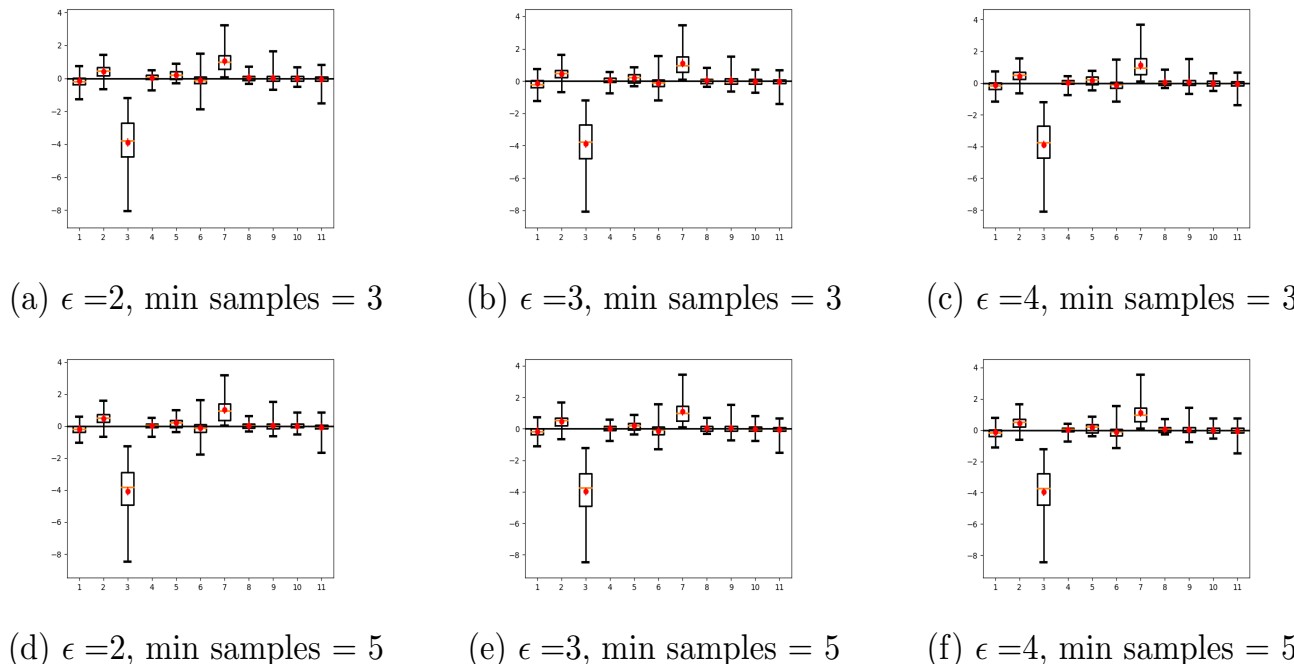

(a) $\epsilon$ =2, min samples = 3  (b) $\epsilon$ =3, min samples = 3  (c) $\epsilon$ =4, min samples = 3

(d) $\epsilon$ =2, min samples = 5  (e) $\epsilon$ =3, min samples = 5  (f) $\epsilon$ =4, min samples = 5

FIG. 3. Estimated regression coefficients (i.e. sensitivity of the model parameters) with median precipitation of the clusters as the response, after clustering with DBSCAN with various parameter values. The red symbols are 95% simultaneous CIs.

the boxplot summary of the empirical sampling distribution and CIs of the effects. Figure 3 shows the sensitivity results when the response is the median (across clusters) of precipitation intensity, and DBSCAN is employed with different parameters. The analogous results for GMM with different values of $NC$ are not shown here, but they are similar. Recall that the variability displayed in each boxplot is due to the 40 days examined. First, note that all of the panels are mostly similar to one another, which implies that the sensitivity results are mostly unaffected by the parameters of the clustering algorithm.

It can also be seen that many of the 11 parameters have a boxplot of values mostly around zero. In other words, when considered across multiple days most of the 11 model parameters have no effect on the median of precipitation, The most obvious exception is

parameter 3, which by virtue of having mostly negative values for its regression coefficient,

is negatively associated with median precipitation. Parameter 7 not only has a weaker

effect (because the median of the corresponding boxplot is closer to zero), it is also not

as statistically significant (because zero falls well within the span of the boxplot). This

parameter is positively associated with precipitation intensity in the typical (median) cluster,

i.e., increasing the parameter leads to more intense clusters; more, below. The conclusions

drawn from an analysis of the CIs in Fig. 3 are the same.

All of these findings are consistent with those found for convective precipitation in

Marzban et al. (2014) where a variance-based sensitivity was performed without any clus-

tering at all. This consistency adds justification to the local/regression-based SA adopted

here, i.e., Eq. (2). It is important to point out that this consistency does not imply that

an object-based SA offers nothing more than traditional non-object-based SA; the former

assesses the sensitivity of object features, something that cannot be done in the latter.

Figure 4 shows the effect of the model parameters on the latitude and longitude of the

clusters (top two rows), amount of precipitation (middle row) in the clusters, and the area

and orientation of the clusters (bottom two rows). The three columns correspond to the

minimum, median, and maximum of a feature. Eccentricity has also been examined, but

the results are not shown here because it is not affected by any of the 11 parameters; this

conclusion is consistent with the results of the omnibus tests performed in the first stage,

mentioned above.

Examination of all of the panels suggests that parameters 4, 5, 8, 9, 10, 11 have little or

no effect on any of the object features. By contrast, parameters 1, 2, 3, 6, and 7 appear to

have varying effects depending on the object feature. Also, the orientation (in addition to

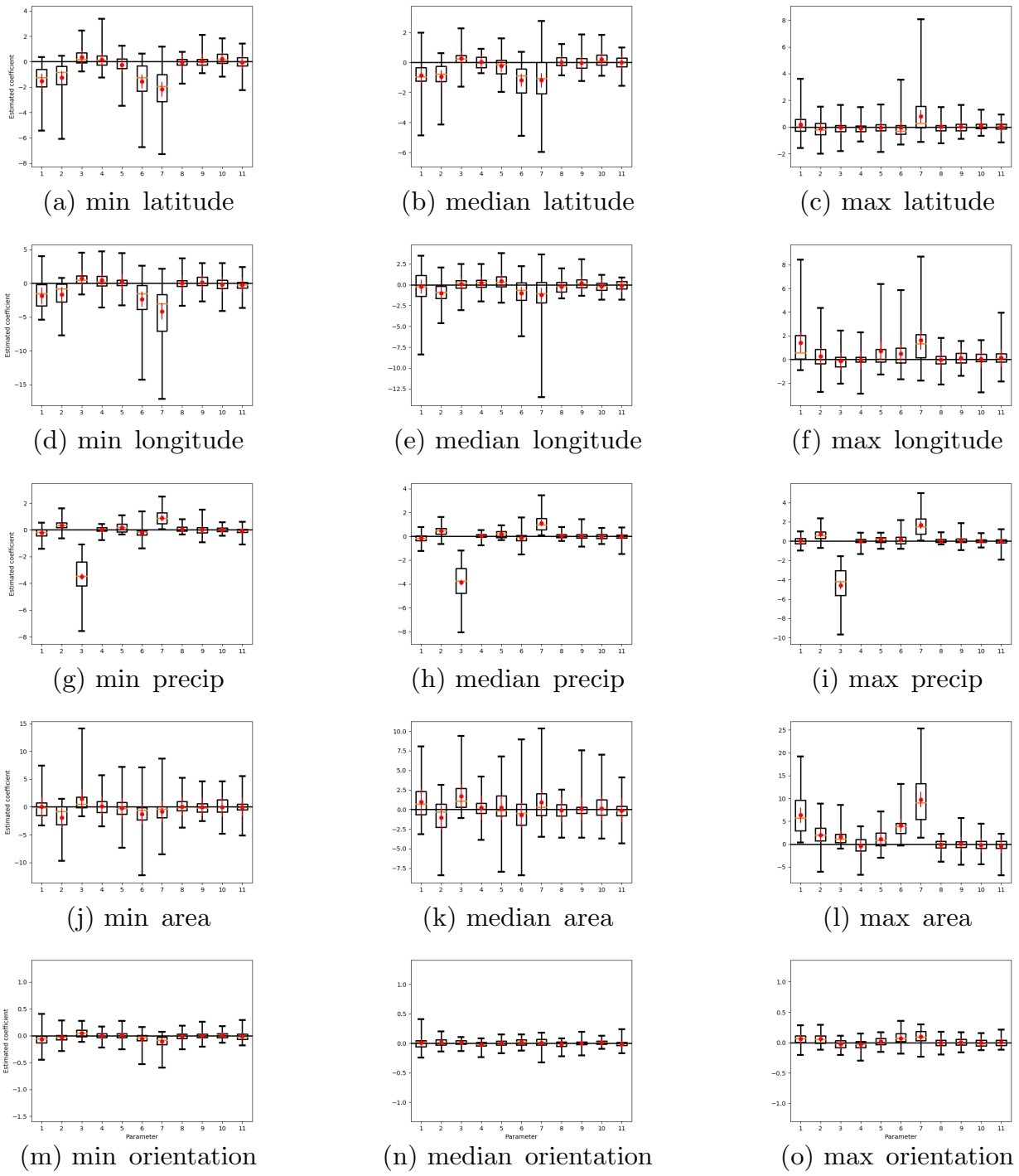

FIG. 4. Estimated MMR coefficients (i.e. sensitivity of the model parameters) on three summary measures (minimum, median, maximum) of different cluster features (latitude, longitude, amount of precipitation, and area and orientation of clusters. Eccentricity is not shown (see text). The red symbols are 95% simultaneous CIs. The clustering is done with DBSCAN with $\epsilon = 2\sqrt{2}$, min_samples = 3.

eccentricity) of the clusters is unaffected by any of the parameters.

The strongest effects are from parameters 3 and 7 on the amount of precipitation. This relationship was already examined in Fig. 3; but now the same pattern can be seen in the minimum, median, and maximum intensity (panels g, h, i in Fig. 4), which implies that the effect of parameters 3 and 7 is to shift down and up, respectively, the whole distribution of precipitation intensity.

The next strongest effects are those of parameters 1 and 7 on maximum area (panel l). Given that these two parameters have no effect on the minimum and median area (panels j and k), it follows that these parameters affect only the right tail of the distribution of size. In other words, by contrast to precipitation intensity whose distribution shifts when parameter 7 is varied, the distribution of size is stretched when that parameter changes. Parameter 6, too, appears to have an effect on maximum area, but to a lesser extent, both statistically and in magnitude.

Whereas parameter 1 tends to stretch out the distribution of area to the right, it appears to have the opposite effect on the minimum and median longitude of the clusters. The effect is weak in magnitude, but statistically significant. It does not affect the maximum longitude (panel f), and so, it stretches the distribution of longitude on the left, causing clusters to appear with smaller longitude, which given the encoding of the data used here, means to the west. Parameters 2, 6, and 7 appear to have the same effect as parameter 1.

The latitude appears to be weakly affected by some of the parameters. For example, parameter 7, and to a much lesser degree parameter 1, is positively associated with median and maximum latitude, but negatively associated with minimum latitude. In other words, increasing parameter 7 increases the width of the distribution of latitude values, causing

$_{514}$ them to be more spread out along the latitudes.

$_{515}$ All of the above conclusions are based on clustering with DBSCAN with $\epsilon = 2\sqrt{2}$ and

$_{516}$ min_samples=3. To test the robustness of these results the same analysis was repeated but

$_{517}$ with GMM as the clustering algorithm and with $NC = 3$. The results (not shown here) are

$_{518}$ mostly the same. One relatively clear difference between the DBSCAN and GMM results is

$_{519}$ in the effect of parameters 1 and 7 on area; whereas with DBSCAN those parameters have

$_{520}$ an effect only on the maximum area, the results based on GMM suggest a significant effect

$_{521}$ on all three distribution summary measures (minimum, median, and maximum area).

$_{522}$ Further differences between DBSCAN and GMM sensitivity results are found when one

$_{523}$ performs a multivariate test for the effect of the model parameters across **all** days. For

$_{524}$ DBSCAN, the p-values corresponding to each of the six cluster features are all found to be

$_{525}$ nearly zero. So, some of the model parameters do have a significant effect on some of the

$_{526}$ features. The same is true for GMM, with the exception of latitude and eccentricity for which

$_{527}$ there is no evidence of an effect (p-values 0.435 and 0.290, respectively). It may appear that

$_{528}$ these results are contradictory, but they are not because the respective parameters of the

$_{529}$ two clustering algorithms have not been tuned to render them comparable. Specifically, the

$_{530}$ DBSCAN parameters are $\epsilon = 2\sqrt{2}$ and min_samples=3, while for GMM the parameter $NC$

$_{531}$ is set to three. In other words, the differences are due to the way in which the two clustering

$_{532}$ algorithms handle their respective parameters. As mentioned earlier, such differences do not

$_{533}$ point to defects in the methodology; they simply reflect the choice of what the user considers

$_{534}$ to be an object.

# 4.  Conclusion and Discussion

It is shown that by employing methods of cluster analysis and sensitivity analysis one can assess the magnitude and statistical significance of the effect of model parameters on the distribution of features (location, intensity, size, and shape) of objects within forecast fields.  For example, one can reveal the model parameters that affect the overall location and/or width of the distribution of object features, and those which impact the shape of the distribution, e.g., by stretching out the left and/or right tail.  The approach does not point to any "optimal" values of the model parameters, for that would require optimizing the model parameters to maximize some measure of agreement between forecasts and observations.  In other words, although the work here lays the foundation for tuning the model parameters for the purpose of improving forecasts in terms of metrics that arise naturally in spatial verification/evaluation methods, no such tuning is performed here.

It is worth pointing out that at least in meteorology, it is not uncommon for different human experts to have different notions of an object in the forecast field.  As such, the ambiguities discussed above are not specific to clustering algorithms, but are inherent to any object-based approach.  In spite of this inherent ambiguity, many spatial verification techniques generally rely on some notion of an object.  The main reason is that accounting for objects in a forecast field is a first step in the verification/evaluation process, and the manner in which objects are defined is of secondary importance.

While this paper is primarily about a methodology, it is worthwhile to provide a possible physical explanation for at least the strongest results in the COAMPS application.  The strongest influence or sensitivity is from parameter 3, the fraction of available precipitation

fed back to the grid from the Kain-Fritsch scheme. Increasing this fraction reduces con-vective precipitation and, based on the results in Marzban et al. (2014), increases stable precipitation, while not affecting total precipitation. It also is responsible for weakening the convective precipitation, i.e., increasing the number of weak systems. The next largest sensitivity is from parameter 7, which controls the temperature difference required to ini-tiate convective precipitation. Again, as shown in Marzban et al. (2014), this parameter also controls a trade-off between convective and stable precipitation and has little effect on total precipitation (along with parameter 1). Parameters 1 and 7 do increase the area of convective precipitation in large precipitation events but not in smaller (areal) precipitation events, likely due to the trade-off between stable and convective precipitation in large events such as frontal systems and mesoscale clusters. This process may also explain the apparent increase in east-west areal coverage and the intensification of precipitation events, as found here.

Several generalizations of the proposed methodology are possible. In Marzban et al. (2008) it has been shown that clustering can be done not only in the 2-dimensional space of latitude and longitude of each grid point, but also in the 3-dimensional space that includes the amount of precipitation at each grid point. In fact, one may argue that the inclusion of more meteorological quantities in the clustering phase ought to lead to more meteorolog-ically relevant objects being identified. In turn, this is more likely to lead to more realistic representation of the effect of the parameters on the object features. The object features may also be extended or revised. For example, here the shape of an object is approximated by an ellipse. But it is possible to use more sophisticated methods of shape analysis (Book-stein 1991; Lack et al. 2010; Micheas et al. 2007; Lakshmanan et al. 2009) to model more

complex shapes. Another possible generalization is to allow for interactions between model parameters. Although the statistical model used here does account for covariance between the model parameters, and between the response variables, no explicit interaction is introduced. The inclusion of such terms is straightforward, and is unlikely to lead to overfitting, at least in linear models such as MMR.

The use of boxplots (in the second stage) to visually display the daily variability of the results is necessarily qualitative. But the authors believe that the information provided in the visual display compensates for the lack of rigor accompanying p-values. CIs are more rigorous than the boxplots, but as mentioned previously, that rigor is accompanied by loss of some information. However, if even more rigor is called for, then it is possible to revise the displays accordingly. For example, one option would be to include a Day factor in the MMR model, and then test the model parameters. Although, the daily variability of the $\beta$ coefficients will be lost, each model parameter will be accompanied by a p-value. Alternatively, one may compute a Bayesian intervals (Leonard and Hsu 1999); such intervals are not necessarily symmetric, and therefore, will be able to convey information on the shape of the underlying sampling distribution. However, they do require additional information, e.g., some knowledge of the prior distribution of the $\beta$'s. All of these options will render the analysis more quantitative, although with a different focus than that emphasized here.[2]

---

[2]The authors acknowledge an anonymous reviewer for these alternatives.

## 5. Code and/or data availability

The code and the data analyzed here occupy about 4.0G of computer space, and are available upon request from the corresponding author, or from https://doi.org/10.5281/zenodo.1043542

## 6. Competing Interests

The authors declare that they have no conflict of interest

## 7. Acknowledgments

This work has received support from Office of Naval Research (N00014-12-G-0078 task 29) and National Science Foundation (AGS-1402895). The authors are grateful to James D. Doyle and Nicholas C. Lederer for providing invaluable support. Ethan P. Marzban is acknowledged for making the flowchart in Figure 1.

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

| ID | Name (Unit) | Description | Default | Range |
|---|---|---|---|---|
| 1 | delt2KF ($°C$) | Temperature increment at the LCL for KF trigger | 0 | -2, 2 |
| 2 | cloudrad ($m$) | Cloud radius factor in KF | 1500 | 500, 3000 |
| 3 | prcpfrac | Fraction of available precipitation in KF, fed back to the grid scale | 0.5 | 0, 1 |
| 4 | mixlen | Linear factor that multiplies the mixing length within the PBL | 1.0 | 0.5, 1.5 |
| 5 | sfcflx | Linear factor that modifies the surface fluxes | 1.0 | 0.5, 1.5 |
| 6 | wfctKF | Linear factor for the vertical velocity (grid scale) used by KF trigger | 1.0 | 0.5, 1.5 |
| 7 | delt1KF ($°C$) | Another method to perturb the temperature at the LCL in KF | 0 | -2, 2 |
| 8 | autocon1 ($\frac{kg}{m^3 s}$) | Autoconversion factors for the microphysics | 0.001 | 1e-4, 1e-2 |
| 9 | autocon2 ($\frac{kg}{m^3 s}$) | Autoconversion factors for the microphysics | 4e-4 | 4e-5, 4e-3 |
| 10 | rainsi ($\frac{1}{m}$) | Microphysics slope intercept parameter for rain | 8.0e6 | 8.0e5, 8.0e7 |
| 11 | snowsi ($\frac{1}{m}$) | Microphsyics slope intercept parameter for snow | 2.0e7 | 2.0e6, 2.0e8 |

KF = Kain-Fritsch, PBL = Planetary Boundary Layer, LCL = Lifted Condensation Level

TABLE 1. The 11 parameters studied in this paper. Also shown are the default values, and the range over which they are varied.