# Peer review of "On the Effect of Model Parameters on Forecast Objects"

_Geoscientific Model Development, 2017_

## Referee Comment (RC1) · Anonymous Referee #1 · 9 Dec 2017

General comments: The paper has been well presented. It has been mentioned that the paper introduces a novel framework. However, it seems that the current authors have used existing methodologies of clustering/statistical analysis that have already been applied to similar problems. Clustering of object fields is a well researched area of study. Therefore, it is not clear how this work adds to the literature. Also, the paper does not provide any specific guidelines on the choice of algorithms and leaves the reader with an ambiguous mind. With these changes, the paper may become a good guidance paper for sensitivity studies.

Specific comments: 1. Please elaborate the abstract to cover some key contributions of the paper or a summary of results in 1-2 sentences. It is incomplete to get the idea of the paper in the current form. 2. At the end of introduction, add more details of the

work done in this paper. Also, add an outline of the various sections that follow. 3. If the goal of the paper is to introduce a novel framework, a flowchart of steps involved in the methods section would be useful.

Technical corrections: 1. Figure 1, 2 and 3 can be made larger and Figure 2 and 3 can spread on the full page. The barplots are hardly visible. 2. Line 78 Provide 1-2 sentence information of COAMPS. What kind of model is it ? 3. Line 115- Does this mean that each model parameter had 9 values associated with it.

---

## Referee Comment (RC2) · Anonymous Referee #2 · 13 Dec 2017

The authors have provided a clearly-written paper demonstrating a method of sensitivity analysis on "objects" in model output. The method is applied and evaluated in an appropriate manner. The description of the method is sufficient to allow for interested scientists to reproduce the results. The sensitivity analysis approach focuses on specific subsets of the model output, namely clusters that have been identified as "objects." The manuscript provides readers with general guidance on determining the sensitivity of various model parameters to relevant aspects of these objects, such as location, amplitude, and orientation. Sensitivity to details in selection of cluster analysis methods was explored, and in the particular example provided, results were shown to not be sensitive to these details.

However, it is not clear that this approach is significantly different from established sen-

sitivity analysis methods. The SA practitioner has to select a "response variable" which is typically a statistic based upon a subset of the full model output. Here, the authors use different cluster analysis approaches to define that subset and various statistics to summarize the model output from that subset. They do not argue for any specific cluster analysis method or statistic, and mention that clusters identified subjectively could also be used. This sounds like traditional SA using subjectively selected subsets of model output, therefore, it is not clear that this is a novel/new approach. Since the results in this manuscript were found to be consistent with previous sensitivity analysis work (Marzban et al. 2014) that did not use objects, it is also not clear that there are significant benefits to using the object-based approach described here. This leads the reader to question the value of going through the extra effort of object segmentation for sensitivity analysis versus traditional SA approaches. It is also not clear if this method has general relevance to the geo-scientific model development community beyond the weather/precipitation prediction application presented here. What other kinds of "objects" could be analyzed in other types of models? I cannot recommend acceptance for publication unless the authors provide a convincing argument for the novelty of the method and provide evidence of the benefits of performing sensitivity analysis on objects in model output.

---

## Referee Comment (RC3) · Anonymous Referee #3 · 20 Dec 2017

*General comments*

In this paper, the authors develop a framework for conducting sensitivity analysis (SA) for the output of numerical models, when the output of interest is a spatial field or realization of a non-smooth or non-continuous variable. In this case, the authors propose conducting the SA on features of "objects," which can be specified generally but here correspond to high quantile clusters of grid cell values of daily precipitation. Two statistical methods are used to determine clusters, namely Gaussian Mixture Model (GMM) clustering and Density-Based Spatial Clustering of Applications with Noise (DBSCAN), and the authors explore sensitivity of their results to the clustering method. Features of these clusters are extracted and fed into a Multivariate Multiple Regression (MMR) model to estimate the effect of individual model parameters on the cluster features.

[Figure]
This is an interesting paper, and describes methodology for an important problem in sensitivity analysis, namely conducting SA for a spatial field of model responses. Furthermore, the clustering both addresses the nature of precipitation data (i.e., non-smooth or non-continuous data) while also reducing the dimension of the problem (i.e., considering fixed clusters rather than a fine spatial grid). The paper is well-written and nicely motivates the work, however, additional detail should be given in the section describing the statistical model, and I think reorganization of Section 3 would greatly improve the presentation (see the Technical corrections below). Furthermore, I am concerned with the analysis methods, particularly the significance testing, and am worried that the way in which the results are presented might be misleading (see the Scientific comments for more details).

*Scientific comments*

As a statistician, I will primarily comment on the statistical model and significance testing, leaving discussion on the experimental design of the sensitivity analysis and variables selected for analysis (i.e., latitude, longitude, intensity, area, orientation, and eccentricity) to more informed parties. In my opinion, the clustering approaches considered (GMM and DBSCAN) seem reasonable, and it was nice to see that results are robust to the clustering method used.

My first concern has to do with the description of the MMR model as well as the treatment of daily replicates within this model. The authors present a generic description of a multiple linear regression model in Equation (1), but it would be helpful to more clearly describe the generalization to the **multivariate** multiple linear regression model that was actually used. If I am following everything correctly, the statistical model you actually use is

$$
\begin{bmatrix} y_t^{min} \\ y_t^{med} \\ y_t^{max} \end{bmatrix} = \begin{bmatrix} \alpha^{min} \\ \alpha^{med} \\ \alpha^{max} \end{bmatrix} + \begin{bmatrix} \beta_1^{min} \\ \beta_1^{med} \\ \beta_1^{max} \end{bmatrix} x_{1,t} + \cdots + \begin{bmatrix} \beta_{11}^{min} \\ \beta_{11}^{med} \\ \beta_{11}^{max} \end{bmatrix} x_{11,t} + \begin{bmatrix} \delta_t^{min} \\ \delta_t^{med} \\ \delta_t^{max} \end{bmatrix}
$$

(where min = minimum, med = median, and max = maximum), or, written in vector form,

$$y_t = \boldsymbol{\alpha} + \boldsymbol{\beta}_1 x_{1,t} + \cdots + \boldsymbol{\beta}_{11} x_{11,t} + \boldsymbol{\delta}_t, \tag{1}$$

for $t = 1, \ldots, 99$ samples taken from the 11-dimensional parameter space. Presumably, you use the usual MMR assumption that the error vectors $\boldsymbol{\delta}_t$ are independent and identically distributed as Normal with mean vector $0$ and non-diagonal covariance matrix $\Sigma$ (i.e., the elements of $\boldsymbol{\delta}_t$ are correlated). Is this a correct characterization of the model?

In practice, you actually estimate the $3 \times 11$ $\beta$ coefficients from Equation (1) for each of six features and each of 40 days, presenting boxplots of the $\beta$ coefficients aggregated over the 40 daily replicates for each of the $3 \times 11 \times 6$ combinations of feature summaries/input variables/features. (See below for a concern related to the boxplots.) This seems like an unnecessary complication to the analysis. As evidenced by your decision to keep only every third day (reducing your data from 120 days to 40 days) in order to remove temporal correlation, it seems to me that these 40 days could represent an ensemble of realizations for each of the 99 parameter settings. Thus, instead of fitting 40 separate MMR models for each of the 6 features, your model could instead be

$$y_{td} = \boldsymbol{\alpha} + \boldsymbol{\beta}_1 x_{1,t,d} + \cdots + \boldsymbol{\beta}_{11} x_{11,t,d} + \boldsymbol{\delta}_{td} \tag{2}$$

for $d = 1, \ldots, 40$ days (I assume that $x_{j,t,d} = x_{j,t}$ for $j = 1, \ldots, 11$, i.e., that the input parameter settings are the same for each day). In other words, instead of using the 40 daily replicates to estimate the distribution of each $\beta$ coefficient, you could build this variation into the statistical model and directly estimate the variability of the coefficients, then calculating $P$-values or confidence intervals as required. This seems to be a more refined way to handle the daily replicates, especially since it seems that you are not concerned with how the $\beta$ coefficients vary across the different days.

Secondly, I am concerned by the significance testing procedure and the presentation of results. First of all, your two-stage procedure for controlling Type I error seems ad

hoc, particularly your qualitative approach to assessing individual significance in the second stage. The omnibus test in the first stage is a good idea (although it would be helpful to have more details given on exactly what you have done – instead of simply providing citations), but you need to be careful about the multiple testing even after reducing the number of tests to $6 \times 40 = 240$. I appreciate that you have at least considered a Bonferroni adjustment, but you should think carefully about this choice: Bonferroni controls a family-wise error rate, implying that the collective conclusion of all tests is invalid if at least one Type I error is made. I don't think this is actually what you want – it seems to me that you simply want to control the number of Type I errors. As an alternative, you might consider the very simple procedure for controlling the rate of false discoveries (i.e., FDR) given in the classic paper by Benjamini and Hochberg (1995). Their simple procedure is remarkably powerful and could more appropriately address the multiple testing issue.

Regardless, after you have conducted the omnibus test, you proceed to present box plots of the coefficient estimates, aggregated across the daily replicates. I think that such an aggregation of the coefficient estimates provides you with a sampling distribution of the true coefficient estimate – please correct me if this is not the right way to think about this. In any case, the aggregated coefficient estimates are most certainly not a posterior distribution of the true coefficient, which is what you would get from a fully Bayesian analysis. In this case, it is misleading to represent a sampling distribution with a boxplot: if the boxplot is skewed to the right, this does not mean that the distribution of the true coefficient is skewed to the right. Instead, you should represent sampling distributions using a confidence interval, which could be plotted as a box (with no whiskers) or a solid bar. Additionally, simply checking to see if boxplots overlap with zero is not an appropriate way to assess *statistical* significance: what significance level is being considered?

My suggestion would be to fold the daily replicates into the MMR as suggested in Equation (2), and calculate $P$-values for each of the $11 \times 3 \times 6$ coefficients. Then,

I would use the Benjamini and Hochberg procedure to identify statistically significant coefficients at a particular level $\alpha$. Instead of the boxplots in Figures 2 and 3, I would recommend using points or bars to indicate the magnitude of the coefficient estimate and shading or masking to indicate which estimates are statistically significant.

*Technical corrections*

On a more technical note, I found the organization of Section 3 to be very confusing. I would suggest moving Sections 3(d) and 3(e) to immediately follow Section 3(a). In this case you will have already described the clustering and the features of interest before discussing the statistical model and significance testing. I would also recommend moving lines 161-170 into Section 3(e).

---

## Author Response (AR1)

Dear Reviewer 1,

Thank you for the review. The following contains your initial review (denoted by ">>"), followed by our immediate reply (denoted by ">"), and our final response (in italic).

>> It has been mentioned that the paper introduces a novel framework. However, it seems that the current authors have used existing methodologies of clustering/statistical analysis that have already been applied to similar problems. Clustering of object fields is a well researched area of study. Therefore, it is not clear how this work adds to the literature.

> It is true that all of the components of the proposed methodology are well-established (to varying degrees); but to our knowledge an object-based sensitivity analysis method has not been developed previously, and certainly not with the specific methods employed by us. More specifically, methods such as 1) clustering, 2) regression models, and 3) sampling methods from experimental design, have not been used together to perform sensitivity analysis of objects in a forecast/spatial field with respect to model parameters. Perhaps it is more accurate to describe our work as a general approach, employing existing methods, for addressing the question of how model parameters affect objects in a forecast field. Again, to our knowledge, no such framework exists, and in that sense the proposed approach is novel.

*We have revised the paper to highlight the importance of each of these main components. A summary of these components is presented in section 2f where the associated problems and our solutions to them are also reiterated.*

>> Also, the paper does not provide any specific guidelines on the choice of algorithms and leaves the reader with an ambiguous mind.

> Our initial intention was to develop a broad framework that can be utilized in a wide range of applications. But it is possible that we have gone too far. As such, we will be happy to add another section in which we provide the reader with some general (but more specific) guidance.

*Section 2b now provides a wealth of guidance on clustering algorithms; in particular, lines 169-233.*

>> With these changes, the paper may become a good guidance paper for sensitivity studies.

> Thank you.

>> Specific comments:

>> 1. Please elaborate the abstract to cover some key contributions of the paper or a summary of results in 1-2 sentences. It is incomplete to get the idea of the paper in the current form.

> Agreed. We will do so.

*Done.*

>> 2. At the end of introduction, add more details of the work done in this paper. Also, add an outline of the various sections that follow.

> Agreed. We will do so.

*Done.*

>> 3. If the goal of the paper is to introduce a novel framework, a flowchart of steps involved in the methods section would be useful.

> Another excellent idea. We will do so.

*Done (Figure 1).*

*In summary, we have added significant material to the paper in order to highlight the novelty of the work (short of using the word "novel!"). The very notion of an object-based SA is novel, and as argued in the paper, there are numerous arenas that may benefit from such a methodology. The clustering sections have also been expanded to provide general guidance to the prospective user.*

Thank you,
Authors

Dear Reviewer 2,

Thank you for the review. The following contains your initial review (denoted by ">>"), followed by our immediate reply (denoted by ">"), and our final response (in italic).

>> However, it is not clear that this approach is significantly different from established sensitivity analysis methods. The SA practitioner has to select a "response variable" which is typically a statistic based upon a subset of the full model output. Here, the authors use different cluster analysis approaches to define that subset and various statistics to summarize the model output from that subset. They do not argue for any specific cluster analysis method or statistic, and mention that clusters identified subjectively could also be used. This sounds like traditional SA using subjectively selected subsets of model output, therefore, it is not clear that this is a novel/new approach.

> The proposed object-based SA is a great deal more than a simple application of traditional SA to a clustered field. In attempting to perform an object-based SA, the SA practitioner will be faced with numerous technical problems whose solutions form the foundation of our proposed methodology. To make that point more clear, we propose to include some version of the following discussion in the paper. It highlights the methodology's novel ingredients, the accompanying problems, and our solutions to them.

> 1) Clustering, as a method for objectively identifying the objects of interest, is a relatively obvious approach. However, it is important for the SA practitioner to be aware that there are at least two distinct ways in which objects can be defined in clustering algorithms, based on a) the number of clusters, and b) the size and distance between clusters. GMM and DBSCAN are the two methods that we have chosen to represent those two approaches.

> 2) Selecting features of the objects, too, may seem straightforward. However, it is not at all obvious that the features can be derived from the covariance matrix. In fact, our initial attempt involved "fitting" closed curves to the objects, a task which is considerably more complicated. In the covariance-based feature selection approach, although we extracted only the simplest of features, there exists a large body of literature which can be of great utility to an SA practitioner.

> 3) Assessing the distribution of each feature presents a more complete picture of the underlying sensitivities than point estimates. The use of multivariate regression (with multiple responses) is a novel (and non-obvious) solution to the problem of summarizing that distribution.

> 4) In a statistical approach to SA, it is important to display both the strength and the statistical significance of the sensitivities. A p-value measures only the latter. The use of boxplots, and the accompanying interpretation we provide, effectively accomplishes both tasks (with some trade-offs, of course).

> Once again, it is true that each of these ingredients, and even the very notion of an object-based SA, could be (re-)discovered by an SA practitioner; what we have described in our paper is the lessons that we have learned from tackling that problem. We believe all of these lessons will be useful for the GMD readership.

*A rephrased version of the above four items is now included in summary section 2f.*

>> Since the results in this manuscript were found to be consistent with previous sensitivity analysis work (Marzban et al. 2014) that did not use objects, it is also not clear that there are significant benefits to using the object-based approach described here.

> It is true that our proposed method, when *specialized* to a "non-object" (e.g. the mean of a field), reproduces results that are consistent with traditional SA results. However, none of our object-based results can be obtained without the object-based SA. In other words, the object-based approach allows one to address questions that a non-object-based approach cannot.

*The following sentence has been added (lines 477-480) to make this clear. "It is important to point out that this consistency does not imply that an object-based SA offers nothing more than traditional, non-object-based SA; the*

*former assesses the sensitivity of object features, something that cannot be done in the latter."*

>> This leads the reader to question the value of going through the extra effort of object segmentation for sensitivity analysis versus traditional SA approaches.

> The reference to "extra effort" suggests that the reviewer may have in mind a situation where the user has an option of choosing between an object-based SA and a non-object-based one. In reality, there is no such option; if the problem at hand calls for SA of object features, then the object-based approach is the only choice; and the "extra effort" is not extra, but necessary.

*The aforementioned sentence (on lines 477-480) addresses this response as well.*

>> It is also not clear if this method has general relevance to the geo-scientific model development community beyond the weather/precipitation prediction application presented here. What other kinds of "objects" could be analyzed in other types of models?

> "Objects" are ubiquitous in Earth Systems. In addition to the meteorology example discussed in the paper, objects arise in models of the ocean (warm/cold eddies, convective plumes, oil spills, ocean garbage transport), volcanic plumes, planet interior, sea ice, vegetation growth, forest fires, and more.

*Additional references have now been included in the paper for some of these examples where objects arise naturally. A figure from each citation has been provided here for the Reviewer's convenience. Objects are evident in all of them, and the features of these objects (e.g., number, size, shape) are all determined by parameters of the underlying models.*

[Figure]

Ocean Eddies

Ocean Garbage

Atmospheric Plume/dispersion

Forest Fires

The Mantle

>> I cannot recommend acceptance for publication unless the authors provide a convincing argument for the novelty of the method and provide evidence of the benefits of performing sensitivity analysis on objects in model output.

> We hope to have presented sufficient arguments to change the Reviewer's opinion.

*In summary, at least to our knowledge, the very notion of an object-based SA is novel, and as we have now argued, there is clearly a need for it in a wide range of fields. Although the development of such a methodology may appear to be straightforward, there are numerous technical problems that must be overcome. Our paper identifies some of these problems, and offers solutions. Although the solutions involve well-established ideas (e.g., Latin hypercube sampling, clustering, multivariate multiple regression, multiple hypothesis testing), these ingredients have not been previously employed for an object-based sensitivity analysis (again, to our knowledge). As such, we believe the work as a whole is sufficiently novel to be considered categorically novel.*

Thank you,
Authors.

Dear Reviewer 3,

Thank you for the review. The following contains your initial review (denoted by ">>"), followed by our immediate reply (denoted by ">"), and our final response (in italic).

>> General comments and Summary …
>> This is an interesting paper, and describes methodology for an important problem in sensitivity analysis, namely conducting SA for a spatial field of model responses. Furthermore, the clustering both addresses the nature of precipitation data (i.e., non-smooth or non-continuous data) while also reducing the dimension of the problem (i.e., considering fixed clusters rather than a fine spatial grid). The paper is well-written and nicely motivates the work, however, additional detail should be given in the section describing the statistical model, and I think reorganization of Section 3 would greatly improve the presentation (see the Technical corrections below). Furthermore, I am concerned with the analysis methods, particularly the significance testing, and am worried that the way in which the results are presented might be misleading (see the Scientific comments for more details).

> We agree with all of your general comments. See more detailed responses below.

>> Scientific comments
>> As a statistician, I will primarily comment on the statistical model and significance testing, leaving discussion on the experimental design of the sensitivity analysis and variables selected for analysis (i.e., latitude, longitude, intensity, area, orientation, and eccentricity) to more informed parties. In my opinion, the clustering approaches considered (GMM and DBSCAN) seem reasonable, and it was nice to see that results are robust to the clustering method used.

> Agreed.

>> My first concern has to do with the description of the MMR model as well as the treatment of daily replicates within this model. The authors present a generic description of a multiple linear regression model in Equation (1), but it would be helpful to more clearly describe the generalization to the multivariate multiple linear regression model that was actually used. If I am following everything correctly, the statistical model you actually use is

     Eqn for MMR with 3 responses

(where min = minimum, med = median, and max = maximum), or, written in vector form,

     Eqn for MRR in vector form        (1)

for $t = 1, \ldots, 99$ samples taken from the 11-dimensional parameter space. Presumably, you use the usual MMR assumption that the error vectors delta_t are independent and identically distributed as Normal with mean vector 0 and non-diagonal covariance matrix Sigma (i.e., the elements of delta_t are correlated). Is this a correct characterization of the model?

> The Reviewer's description of our model is correct, and we will be happy to include the additional details in the paper.

*We have now revised the description of MMR to align it with the Reviewer's presentation.*

>> In practice, you actually estimate the 3 X 11 beta coefficients from Equation (1) for each of six features and each of 40 days, presenting boxplots of the beta coefficients aggregated over the 40 daily replicates for each of the 3 X 11 X 6 combinations of feature summaries/input variables/features. (See below for a concern related to the boxplots.) This seems like an unnecessary complication to the analysis. As evidenced by your decision to keep only every third day (reducing your data from 120 days to 40 days) in order to remove temporal correlation, it seems to me that these 40 days could represent an ensemble of realizations for each of the 99 parameter settings. Thus, instead of fitting 40 separate MMR models for each of the 6 features, your model could instead be

     Eqn for MMR in vector form but across all days     (2)

for $d = 1, \ldots, 40$ days (I assume that x_jtd = x_jt for $j = 1, \ldots, 11$, i.e., that the input parameter settings are the same for each day). In other words, instead of using the 40 daily replicates to estimate the distribution of each beta coefficient, you could build this variation into the statistical model and directly estimate the variability of the coefficients, then calculating P-values or confidence intervals as required. This seems to be a more refined way to handle the daily

replicates, especially since it seems that you are not concerned with how the beta coefficients vary across the different days.

> Here the Reviewer is concerned over how daily variability is "handled." In the paper, we developed an MMR for each of the 40 days, while the proposed model in Eq (2) above, would "average" over the daily variability. While the latter model may make sense from the perspective of a Statistician aiming to build a most parsimonious model, the fact is that in most SA applications daily variability is something that users want to see. As such, averaging over it is not desirable for practitioners. There is a third alternative - introducing a factor, denoted Day, on the right side of the model. In other words, in the language of experimental design, one can block the Day factor. We have actually performed that analysis as well. There are pros and cons to that work.
> In general, on the one hand, blocking the Day factor is expected to make it easier to detect a statistically significant effect in the other 11 beta coefficients (i.e., it can increase power). On the other hand, because of the restriction on randomization (hence, treating Day as block), one cannot rely on the tests of significance for a block (i.e., Day) effect. Even if one were to believe the p-value associated with the Day factor, it would be only one number! And that brings us back to what we said earlier, namely that in most applications users desire to see the daily variability.
> Now, that is all generalities and expectations; but what about the problem at hand? As we said, we have actually done the analysis of including the Day factor in the model as a block. Some of the results are reasonable conclusive. For example, when the response is simply the domain average of the forecast (i.e., not object-based at all), we found that blocking the Day factor has no effect on the estimates of the other 11 beta coefficients. But when dealing with objects the results do not suggest any simple conclusion! For that reason, we decided to exclude it from the paper. However, if the Reviewer believes this is too important to ignore, we will be happy to discuss it (perhaps in an appendix, in order to not disrupt the flow of the paper).

*First, contrary to the Reviewer's initial assessment that we "are not concerned with how the beta coefficients vary across the different days," we actually do care about the daily variability of the betas. We had mentioned this in the earliest version of the paper, but we have now reiterated it in many places. Also, as mentioned previously, we have the results of the analysis wherein a Day factor is included in the model. Note that the current results (e.g., each panel in Figure 5) involves 11 boxplots, while those from the model that includes a Day factor involve 11 p-values. We have confirmed that the p-values are consistent with the boxplots; small p-values are associated with boxplots that are far from the zero line, and large p-values correspond to boxplots that have a significant overlap with the zero line. But, as we expressed in our initial response, it is immediately evident that the 11 boxplots carry a lot more information than 11 p-values. Consequently, although we agree with the Reviewer in that from a statistician's perspective it makes good sense to include a Day factor in the model, given the importance of viewing daily variability, we have opted for the boxplots. However, as per the Reviewer's suggestion we have produced confidence intervals, superimposed on the boxplots. A brief comparison of the pros and cons of boxplots and confidence intervals is given on lines 370-385.*

>> Secondly, I am concerned by the significance testing procedure and the presentation of results. First of all, your two-stage procedure for controlling Type I error seems ad hoc, particularly your qualitative approach to assessing individual significance in the second stage.

> We are surprised by the Reviewer's opinion on the 2-stage procedure. Outside of the multiple-hypothesis-testing circles, it is *the* approach to testing. One begins with a single omnibus test, and only if it's rejected one proceeds to performing multiple tests. There are numerous articles advocating the wisdom in this practice, and we will be happy to include them in the paper.

*We maintain that the two-stage approach in linear models is a standard and time-tested procedure, and we have provided references and more explanation of omnibus tests to support that belief. We have also demonstrated the usefulness of the 2-stage procedure through an example. Specifically, in the first stage of the procedure, i.e., without examining the effects of each model parameter on each response separately, there is already evidence that eccentricity is not affected by any of the model parameters. As such, there is no reason to perform multiple hypothesis testing of the effect of each model parameter on each response. The omnibus test in the first stage is not intended to control errors associated with multiple hypothesis testing involving model parameters and responses; it is designed to avoid that testing altogether. Abandoning the 2-stage procedure is tantamount to ignoring the utility of omnibus tests in linear*

*models. Now, because there is still multiplicity across the 40 days and 6 features, we have implemented FWER- and FDR-controlling procedures.*

>> The omnibus test in the first stage is a good idea (although it would be helpful to have more details given on exactly what you have done - instead of simply providing citations),

> The omnibus test we performed is an F-test (again, a standard choice). Is this the kind of detail the Reviewer is proposing?

*The omnibus test performed is a generalization of the F-test called Pillai's trace test. This test is now named on line 342.*

>> but you need to be careful about the multiple testing even after reducing the number of tests to 6 X 40 = 240. I appreciate that you have at least considered a Bonferroni adjustment, but you should think carefully about this choice: Bonferroni controls a family-wise error rate, implying that the collective conclusion of all tests is invalid if at least one Type I error is made. I don't think this is actually what you want - it seems to me that you simply want to control the number of Type I errors. As an alternative, you might consider the very simple procedure for controlling the rate of false discoveries (i.e., FDR) given in the classic paper by Benjamini and Hochberg (1995). Their simple procedure is remarkably powerful and could more appropriately address the multiple testing issue.

> It is not clear to us which error rate - FWER or FDR - is more appropriate to control for gridded fields, so we shall report the results of both. However, let us point out that the choice of the controlled error rate has very little bearing on the majority of the results in the paper, because in spite of the prevalence of p-values very little hypothesis testing is actually performed. There are only a few places in the paper where we report counts of significant effects. The remainder of the conclusions are based on the visual assessment of boxplots; in this connection, please see our response below.

*The FWER and FDR are now presented on lines 320-330, and the results are reported on lines 448-455.*

>> Regardless, after you have conducted the omnibus test, you proceed to present box plots of the coefficient estimates, aggregated across the daily replicates. I think that such an aggregation of the coefficient estimates provides you with a sampling distribution of the true coefficient estimate - please correct me if this is not the right way to think about this.

> It may be safer to call it the "empirical" sampling distribution. Even then, some may object to calling it a sampling distribution because sampling across days is hardly a random sample from a population. But, yes, these boxplots are intended to summarize some proxy for the sampling distribution of the respective regression coefficients.

*The revised paper now follows the Reviewer's terminology, referring to the boxplots as providing a 5-point summary of the empirical sampling distribution.*

>> In any case, the aggregated coefficient estimates are most certainly not a posterior distribution of the true coefficient, which is what you would get from a fully Bayesian analysis. In this case, it is misleading to represent a sampling distribution with a boxplot: if the boxplot is skewed to the right, this does not mean that the distribution of the true coefficient is skewed to the right.

> Given that we have no a priori reasons for believing that there should be a skew, we have no reason to choose anything other than a symmetric a priori pdf. As such, the skew in the boxplots does translate to a skew in the posterior pdf.

>> Instead, you should represent sampling distributions using a confidence interval, which could be plotted as a box (with no whiskers) or a solid bar.

> A confidence interval has two "drawbacks:"
1) It does not convey the shape of the underlying distribution - a useful quantity, and
2) It depends on a significance/confidence level (see next comment, below).

*As we mentioned above, in spite of these "drawbacks," confidence intervals are now supplemented to the boxplots.*

>> Additionally, simply checking to see if boxplots overlap with zero is not an appropriate way to assess statistical significance: what significance level is being considered?

> The Reviewer is correct in that boxplots alone are not sufficient for performing hypothesis testing - one also requires some kind of threshold, e.g., significance level.  However, as we have indicated above and in the paper, in spite of the prevalence of p-values in the paper, we actually do very little hypothesis testing (i.e., rejecting/not-rejecting). This is intentional. Although some problems can benefit from a simple significant/not-significant summary of results, in the case of our problem, we believe it is more informative to display the empirical sampling distributions. Although this certainly introduces a subjective/qualitative ingredient into the analysis, we believe that it displays the results in a more holistic manner, and therefore, is a more useful trade-off. This philosophy is in line with the policy that many journals and practitioners are following in that summarizing complex results in terms of a binary reject/no-reject decision, or a p-value, or a confidence interval, leads to loss of information. We are hoping that the Reviewer will see the benefits of this trade-off, but if necessary we are willing to superimpose some sort of confidence interval on the boxplots (or the alternative shaded-point plots proposed below).

*As mentioned above, given that boxplots and confidence intervals represent a different trade-off between the information at hand, we have decided to display both.*

>> My suggestion would be to fold the daily replicates into the MMR as suggested in Equation (2), and calculate P-values for each of the 11 X 3 X 6 coefficients. Then, I would use the Benjamini and Hochberg procedure to identify statistically significant coefficients at a particular level alpha. Instead of the boxplots in Figures 2 and 3, I would recommend using points or bars to indicate the magnitude of the coefficient estimate and shading or masking to indicate which estimates are statistically significant.

> As we have indicated, we are amenable to discussing the various ways in which daily variability can be handled, and reporting counts of significant effects based on both FWER and FDR control. We can also see the benefit of replacing the boxplots with something that shows each of the members in the boxplots. Although this will take some experimentation on our part, we will do it because it's a good idea.

*#All of the above suggestions have now been implemented, with the exception of including a Day factor in the model. To reiterate, the daily variability is a sufficiently important source of variability (at least in meteorology) that it deserves a visual display; including a Day factor in the model, although statistically more rigorous, denies the user that luxury.*

>> Technical corrections
>> On a more technical note, I found the organization of Section 3 to be very confusing.  I would suggest moving Sections 3(d) and 3(e) to immediately follow Section 3(a). In this case you will have already described the clustering and the features of interest before discussing the statistical model and significance testing. I would also recommend moving lines 161-170 into Section 3(e).

> We were aware that there is some "back-and-forthing" in that section, but we believed that structure was a reasonable trade-off. However, if the Reviewer found it "very confusing," then we will be happy to re-organize as suggested.

*The sections have been moved as per the Reviewer's suggestion.*

*In summary, we have responded actively to the Reviewer's suggestion to expand the discussion of MMR, explain omnibus tests, restructure the presentation, include FWER- and FDR-controlling procedures, and supplement the boxplots with confidence intervals. The exceptions are in not including a Day factor in the MMR model (explained in the response denoted #, above), and maintaining the 2-stage nature of the procedure (explained in the response denoted \*).*

Thank you,
Authors.

[revised manuscript text omitted]